# Twin home birth: Outcomes of 100 sets of twins in the care of a single practitioner

**Stuart J. Fischbein**[ID][1], **Rixa Freeze**[ID][2]* *

**1** Birthing Instincts, Kanab, Utah, United States of America, **2** Breech Without Borders, Crawfordsville, Indiana, United States of America

☯ These authors contributed equally to this work.

* rixa@breechwithoutborders.org

## Abstract

### Background

Research on community (home or birth center) twin birth is scarce. This study evaluates outcomes of twin pregnancies entering care with a single community practitioner.

### Methods

This is a retrospective observational cohort study of 100 consecutive twin pregnancies planning community births during a 12-year period. Outcomes measured included mode of birth; birth weights; Apgar scores; ante-, intra-, and post-partum transports; perineal integrity; birth interval; blood loss; chorionicity; weight concordance; and other maternal or neonatal morbidity.

### Results

31 women (31%) transferred to a hospital-based clinician prior to labor. Of the 69 pregnancies still under the obstetrician's care when labor began, 79.7% (n = 55) were Dichorionic Diamniotic and 21.3% (n = 14) were Monochorionic Diamniotic. The vaginal birth rate was 91.3% (n = 63): 77.3% for primips and functional primips (no previous vaginal births) and 97.9% for multips. Six mothers (8.7%) had in-labor cesareans (1 multip and 5 primips). Rates of vaginal birth did not vary significantly by chorionicity. There were 8 transports in labor (11.6%): 2 vaginal and 6 cesareans. Average gestational age was 39.0 weeks (range 35–42). Compared to primiparas, multiparas had less perineal trauma and higher rates of vaginal birth and spontaneous vaginal birth. One twin infant and one mother required postpartum hospital transport. Of the babies born in a community setting, there was no serious morbidity requiring hospital treatment.

### Conclusions

A community birth can lead to high rates of vaginal birth and good outcomes for both mothers and babies in properly selected twin pregnancies. Community twin birth with midwifery style care under specific protocol guidelines and with a skilled practitioner may be a reasonable choice for women wishing to avoid a cesarean section—especially when there is no

**Data Availability Statement:** All relevant data are within the article and its Supporting information files (S4 Spreadsheet).

**Funding:** The author(s) received no specific funding for this work.

**Competing interests:** The authors have declared
that no competing interests exist.

option of a hospital vaginal birth. Training all practitioners in vaginal twin and breech birth
skills remains an imperative.

## Introduction

This paper examines maternal and neonatal outcomes of a series of 100 twin pregnancies sup-
ported by an obstetrician who has been attending home births exclusively since 2010, and
prior to that, 24 years of hospital practice attending breeches, twins, and VBACs (vaginal birth
after cesarean). His decision to attend home births was driven by breech, twin, and VBAC
bans and limited options at his local hospitals. In California, midwives are not allowed to
attend twin births at home or in birth centers, leaving this physician as one of the only options
for a community (home or birth center) birth in the entire state with a population of 39 million
people.

Around 120,000 babies are born as twins every year in the USA (United States of America),
representing 3.2% of all births [1]. As of 2013, the last year in which the data has been pub-
lished, the C-section rate for twins was 74.8%, slightly lower than the 2009 peak of 75.3% [2–
4]. With decreasing skill and experience in vaginal twin births, providers are relying more
heavily on cesarean delivery [5]. However, this presents a difficult situation for women who
strongly desire a vaginal birth. If there are few or no hospital options for their twins, a home
birth may be their only way to avoid unwanted surgery.

American women are increasingly turning to home birth for many reasons including
increased comfort, lower intervention rates, greater feeling of safety, lack of acceptable hospital
options, and recently, concerns over Covid-19 infection and/or Covid regulations at their local
hospitals [6–10]. In the USA, home births rose 19% between 2019 and 2020, largely due to the
Covid-19 pandemic and the associated hospital restrictions [11, 12]. For women with multiple
gestations, hospital options are even more restricted than for those with singleton cephalic
pregnancies. Where a vaginal hospital twin birth is still possible, the process is often medical-
ized (common practices include induction at a set gestational age, vaginal birth in the OR, epi-
durals as mandatory or strongly encouraged, mother in dorsal lithotomy position, and strict
time limits for birth interval). Some women find this medicalized approach unacceptable and
choose to give birth outside the medical system [13, 14]. With ACOG (American College of
Obstetricians and Gynecologists) contraindicating multiple births at home and some state reg-
ulations forbidding community midwives from supporting twin pregnancies, the options for a
vaginal birth decrease even further.

While literature on outcomes of planned home birth is extensive, showing good perinatal
outcomes with high rates of vaginal birth and maternal satisfaction and low rates of interven-
tion [15–19], very little data exist on outcomes of twin births in a community setting (home or
birth center). Obstetrical societies such as the ACOG list twin birth at home as contraindi-
cated, due to (short-term) safety concerns for the neonate. However, systematic, detailed data
are not available for twin home births, particularly data that take into account provider skill
and experience levels, chorionicity, selection criteria, or type of presentations. A 2005 study of
planned home births by CPMs (Certified Professional Midwives) briefly mentions 13 sets of
twins with no neonatal deaths [20]. In the MANA (Midwives Alliance of North America) Stats
2.0 and 4.0 data sets, analyzed in 2017, there were 138 twin pregnancies intending to birth at
home, representing 0.3% of births out of a total of 47,394 women planning home or birth cen-
ter births [21]. This study includes more detailed information on outcomes, such as Apgar

scores, NICU (newborn intensive care unit) stays, and maternal injuries or PPH (postpartum hemorrhage), although it still lacks information on provider skill level, chorionicity, selection criteria, gestational age, or birth weights. Of the twin pregnancies, 19.6% ended in intrapartum transfer, with an overall cesarean rate of 18.1%. Compared to planned singleton home birth, twin pregnancy was not associated with higher rates of neonatal transfer, perineal trauma, or neonatal death. On the other hand, twin pregnancy was associated with higher rates of intrapartum transfer and maternal or neonatal hospitalization ("modest"); of postpartum transfer, cesarean section, low 5-minute Apgars, and NICU admission ("elevated"); and of very low 5-minute Apgars ("substantially elevated")—again, compared to planned singleton home birth.

A small number of studies outside the USA mention twin community births, but with few details about the pregnancies or about the outcomes. Bastian et al. [22] analyzed self-reported data on home births in Australia and reported a perinatal mortality rate of 1:7 for the twin births. This study did not note how many twin pregnancies were included in the data, and the study is now 25 years old. A few other studies looking at the role of "high-risk" home births (including multiple gestations) have found an increased rate of adverse outcomes in the high-risk groups, but the role of twins was not parsed out separately [23–25]. A PhD dissertation examining more than a half-million home and hospital births in the UK SMMIS registry from 1988–2000 found no cases of perinatal mortality among the planned twin home births; there were 13,263 multiple pregnancies overall in the SMMIS registries, of which some (likely a small percentage at most) were planned home births [26].

There is a wide disparity between evidence and practice regarding mode of birth for twin presentations [27, 28]. A large body of hospital-based research indicates that there is no benefit to routine cesarean section for twins, regardless of whether twin A or B is cephalic or breech. The 2013 Twin Birth Study was a randomized controlled trial comparing planned vaginal birth with planned CS for cephalic-first "low-risk" twins, enrolling 2804 women in 106 centers between 32 + 0 and 38 + 6 weeks' gestation. It found no benefit to planned cesarean section [29]. A large prospective observational study of twin birth outcomes in France and Belgium, called JUMODA, found no benefit to cesarean for cephalic first twins [30]. JUMODA was designed with identical exclusion criteria as the Twin Birth Study to enable cross-comparison of a low-risk population. (JUMODA is a sister study to PREMODA, which examined outcomes of singleton breech births in maternity centers in France and Belgium and found no benefit to planned cesarean section for breech presentation.) The Cochrane review on the use of cesarean section for multiple gestations found two only relevant RCTs and concluded, "There is insufficient evidence to support the routine use of planned caesarean section for term twin pregnancy with leading cephalic presentation." The Cochrane review has no recommendation about mode of birth for breech-first twins as no RCTs exist [31]. A sub-analysis of the JUMODA data by Korb [32], looking at breech-first twins, found no benefit to cesarean for those presentation, echoing similar findings by Blickstein, Goldman, and Kupferminc in 2000 [33], Bourtembourg et al. in 2012 [34], and Ghesquière et al. in 2022 [35].

Despite strong evidence that a planned vaginal birth should be offered and supported for most twin presentations, clinical practice in the USA and many other countries strongly favors planned cesarean section. In many places in the USA, women cannot access planned vaginal birth at all, even when both babies are cephalic. This lack of choice is difficult to overcome due to the deskilling of practitioners [32, 36].

Given this medical and cultural context—with evidence supporting vaginal birth for most twin pregnancies, but with clinical practice, hospital protocols, and state regulations making vaginal birth either impossible or untenable for most women—we find it beneficial to analyze outcomes of planned twin births at home or in a birth center with a skilled, experienced

practitioner. In many locations in the USA, community birth may be a woman's only way to achieve a vaginal birth if she has twins, particularly breech-first twins.

## Materials and methods

This paper is a retrospective review of 100 consecutive twin mothers who entered care for a community birth with a single practitioner over a 12-year period from March 2011 to January 2023. Seven VBACs were included in this series; a mother without a previous vaginal birth was considered a functional primipara. The women in this series were all in good health and had no underlying medical complications; they received prenatal care with an obstetrician, a midwife, or a collaboration of both. Most of the twin clients self-selected the option of home birth at an early gestational age; they were highly motivated and experienced continuity of care throughout their pregnancies.

Prenatal visits were individualized based on history, social factors such as distance, client preference, and chorionicity. Dichorionic Diamniotic (Di-Di/DCDA) twins without concerns were seen about every 4 weeks through 28 weeks, then every 2 weeks until 35–36 weeks, and then weekly or as needed. Monochorionic Diamniotic (Mono-Di/MCDA) twins were seen more often in the second trimester for surveillance. This was again individualized but usually every 2 to 3 weeks unless more often was clinically indicated (for example, growth restriction or deviation from their individual growth curves). Ultrasound surveillance was also individualized based on clinical setting and parental desire with recommendations for closer monitoring based on chorionicity and growth.

The women in this review were selected and accepted to labor based on simple criteria as follows:

- Greater than 35 weeks gestation

- Di-Di or Mono-Di chorionicity

- Twin A in a stable longitudinal lie

- Both twins growing consistently on their own ultrasound growth curves with EFW (estimated fetal weight) of smallest twin >2000 gm

- No gross anomalies or significant maternal medical issues (i.e., Type 1 diabetes, heart disease, or serious clotting disorders)

- Spontaneous onset of labor

- Fetal and maternal tolerance to labor

- Right parental mindset for community-based birthing (motivated, confident, and well-informed parents who did not want what the obstetric model offered them)

These were used as general guidelines, not as strict selection criteria. Each case was individualized per our model of care and ethical considerations (such as the right to informed decision making; the right to refuse care; and the right to non-coercive, non-judgmental care). The model of care allowed the team ample time during prenatal visits to provide information on all available options, with their associated risks, benefits, and alternatives.

The birth team consisted of an obstetrician (SJF), two licensed midwives (either direct-entry midwives with a CPM [Certified Professional Midwife] credential or Certified Nurse Midwives), one or two midwifery students, and often a birth doula or additional birth assistants. Duties were discussed ahead of time as to who had primary responsibility for Twin A, Twin B, and the mother. Assistants and students were in charge of setup, cleanup, and

documentation. Equipment brought to each birth included IV fluids and tubing, sterile gloves, gauze, pads, betadine, urinary catheters, suture material, and appropriate surgical instruments. The birth team also supplied an inflatable birth pool. Medications included antibiotics, lidocaine, oxytocin, misoprostol, oral methylergonovine (until it became unavailable), tranexamic acid (TXA), vitamin K, Rhogam, smelling salts, a Masimo pulse oximeter, handheld Doppler, and oxygen. The above equipment and medications are common in many USA home birth practices. In this series SJF also carried a portable GE ultrasound, a Mityvac vacuum, Piper forceps, Simpson forceps, and Tucker-McLean forceps. All licensed practitioners were certified in neonatal resuscitation (NRP or Neonatal Resucitation Program) and cardio-pulmonary resuscitation. Normal resuscitation equipment included an Ambu bag, DeLee suction, and laryngeal mask airway.

Women in this cohort were not excluded for conditions that were unlikely to affect labor such as diet-controlled gestational diabetes, mild chronic hypertension, IVF pregnancy, or age over 35. The data were not analyzed prior to completion of the 100th twin birth.

SJF followed a midwifery style of care, encouraging settings where women feel "private, safe, and undisturbed" to encourage the proper release of birth hormones involved in physiological birth: oxytocin; beta-endorphin; catecholamines (adrenaline and noradrenaline), and prolactin [37]. Under this model of care, women are encouraged to eat and drink, ambulate, change positions, and choose their birth location and position. There is continuity of care and familiarity with the birth team, creating relationships of trust [38, 39]. This model is sometimes referred to as a "low-tech high-touch" approach [40]. This model of care is not exclusive to midwives; it is defined by care practices, not by the practitioner's title. Some midwives practice a more medical model of care, while some obstetricians follow a midwifery model of care.

Breech and cephalic labors were managed identically. Women were encouraged to eat and drink, ambulate, change positions, and choose their birth location and position. Women had the option of a shower and tub for labor analgesia; water birth for breech births was not the preferred mode due the higher likelihood of assistance, but choice remained with the laboring woman. Fetal monitoring was performed intermittently with a Nicolet Elite 200 Handheld Doppler. Auscultation was individualized but the usual protocol practice was every 30–60 min in active labor, every 15–30 min in transition and every 5–10 min in the second stage. Vaginal exams were only done with maternal informed consent and kept to a minimum, often withheld until maternal guttural vocalizations signaled an urge to push. Portable ultrasound was used sparingly mainly to confirm position and heart rate of twin B after birth of twin A.

Pushing only began when maternal urge became irresistible; pushing was spontaneous rather than coached. Passage of pasty meconium was considered a positive sign of descent with a breech. Breech twin mothers were counseled about the benefits of upright and hands-off techniques. On-the-back positioning for breech was used on an as-needed basis and with maternal informed consent. Optimal cord clamping (usually done after the placenta was birthed) and immediate and uninterrupted skin-to-skin were routine.

This project received approval from the University of Michigan IRB (HUM00195378). Patient informed consent to participate was waived by the IRB as this was a retrospective chart review using de-identified data excerpted by SJF into an Excel spreadsheet from the clients' medical records (see S1 Data). The records were accessed for research purposes on April 29, 2022. The data was collected by SJF and then de-identified to remove any personal identifying information prior to analysis. Sample sizes were too small for most statistical analysis. We performed a limited statistical analysis of Apgar scores using unpaired t-tests on GraphPad.

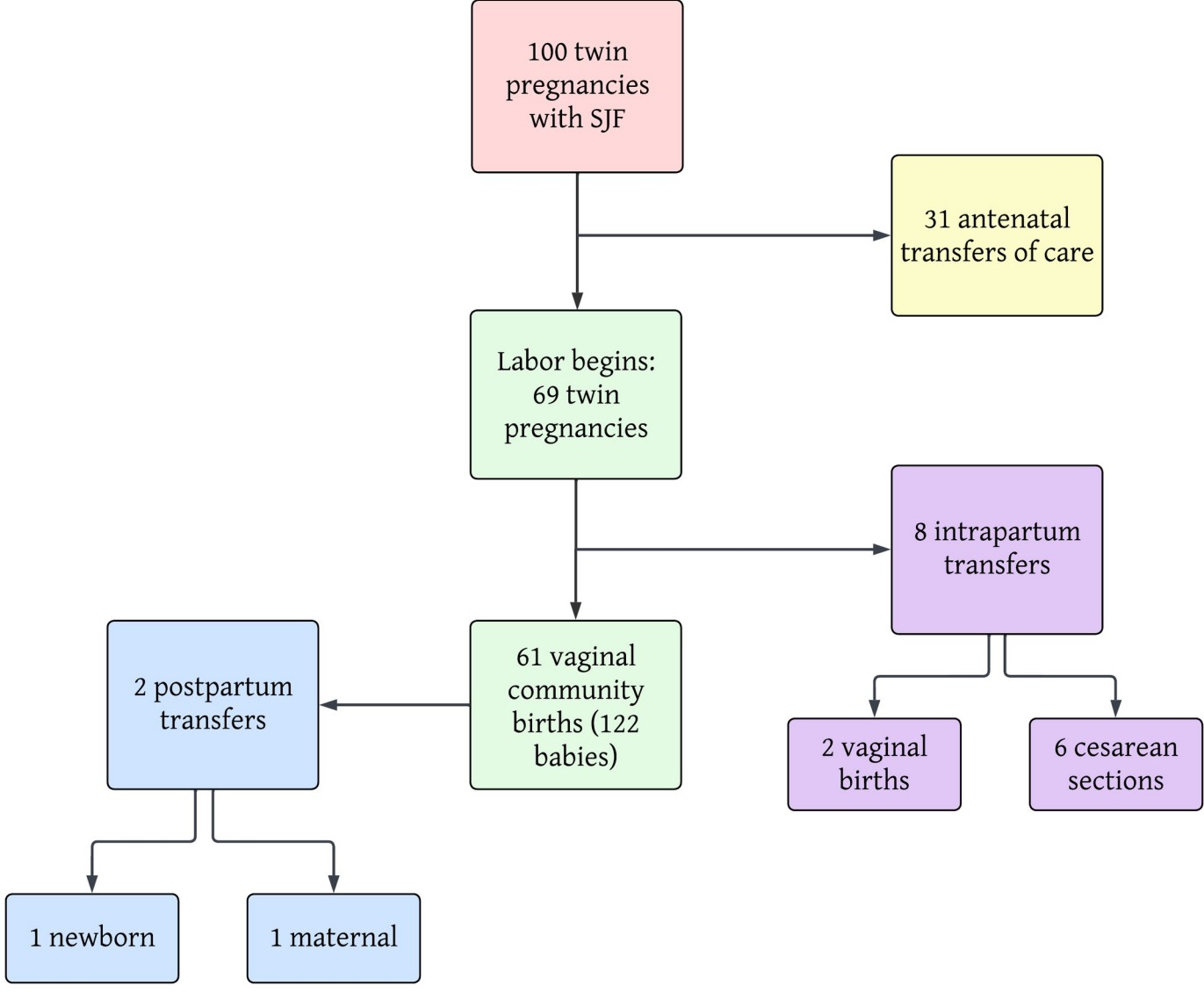

**Fig 1. Flowchart of 100 sets of twins.**

## Results

A total of 100 women planning a community-based birth with either Dichorionic Diamniotic (79%) or Monochorionic Diamniotic (21%) twins received care with a single obstetrician over a twelve-year period from March 2011 to January 2023 (see Fig 1).

Antepartum transfer of care (TOC) occurred with 31 of the women. Of the antepartum women whose care was transferred to a hospital-based practice, 19.4% (6/31) were Mono-Di and 80.6% (25/31) were Di-Di, similar to the overall percentage in the cohort. There were 4 cases of twin-twin transfusion syndrome (TTTS) diagnosed in the Mono-Di twins (19%), all of whom are included in the antepartum group (see Fig 1 and Table 1).

The remaining 69 women went into spontaneous labor. Of these, 8 women (11.6%) were transported in labor, one emergently and 7 non-emergently (see Table 2). Of these transfers, 2 went on to birth vaginally with a supportive hospital practitioner and the other 6 had cesarean

**Table 1. Reasons for antenatal transfer of care to a hospital-based provider.**

| | Primip twins n = 19 | Multip twins n = 12 |
|---|---|---|
| >41 weeks, NIL (not in labor) | 3 | |
| Cholestasis | 2 | |
| SPROM (spontaneous preterm rupture of membranes) < 35 weeks | 2 | 2 |
| Preterm labor < 35 weeks | 3 | 4 |
| Gestational hypertension/preeclampsia | 3 | 1 |
| IUGR (intrauterine growth restriction) | 1 | |
| Funic or footling prior to labor | 2 | 2 |
| TTTS (twin to twin transfusion syndrome) | 2 | 2 |
| Non-reassuring FHT (fetal heart tones), NIL | 1 | |
| Twin A breech (early in SJF's home birth practice) | | 1 |
| Total antenatal transfers | 19/40 (47.5%) | 12/60 (20.0%) |

sections. The overall cesarean section rate in the cohort planning home birth at the onset of labor was 8.7% (5 primips & 1 multip).

Of the remaining 61 clients who birthed at their desired location, there were 15 primiparas (including 5 first-time VBAC moms) and 46 multiparas (including 1 VBAC). Vaginal birth was achieved in 97.9% (46/47) of multiparas and 77.3% (17/22) of primiparas who entered labor, with 68.2% (15/22) of primiparas giving birth vaginally in the community setting (see Fig 2). For a video of a community home birth with SJF, please see S1 File.

For those 69 women who planned a community birth, the presentation of the twins broke down as follows in Table 3 and Fig 3. For more details about the VBAC twin births, please see S2 File.

Of the 61 births that occurred in a community setting, 8 involved vacuum extraction of a vertex baby (7 for baby B only, 1 for both twins) and one involved forceps for a vertex baby B. There were 2 vacuum extractions due to concerns about FHR of the second twin, 2 for cord prolapse of the second twin, 3 for ineffectual pushing, and one for a long birth interval. Forceps were used once to assist a direct OP presentation with low FHR. Of these nine instrumental deliveries, there were three intact perinea, three 1st degree lacerations, two 2nd degree lacerations, and one 3rd-degree laceration (this last one after vacuum for baby A, internal podalic version & breech extraction of baby B), with no episiotomies. Because these births took place in a home setting, all interventions (such as breech maneuvers or extraction, instrumental delivery, or manual removal of the placenta) were done without anesthesia and tolerated well by the mothers.

Close to half of the pregnancies had at least one twin presenting breech at the onset of labor (44.9%, 31/69). Of the 35 total breech presentations, 12 were frank, 19 were complete, 3 were

**Table 2. Indications for in-labor hospital transfer of twins.**

| Indication for transfer | Primip twins (includes 2 VBACs) n = 7 | Multip twins n = 1 |
|---|---|---|
| Arrest of active labor (> 7 cm) | 2 | 1 (B/B) |
| Audible decelerations in early labor | 1 | |
| Maternal complaint of incisional pain (VBAC) | 1 | |
| SPROM, prolonged latent labor, maternal fatigue | 3 | |
| **Total in-labor transfers by parity** | **7/22 (31.8%)** | **1/47 (2.1%)** |

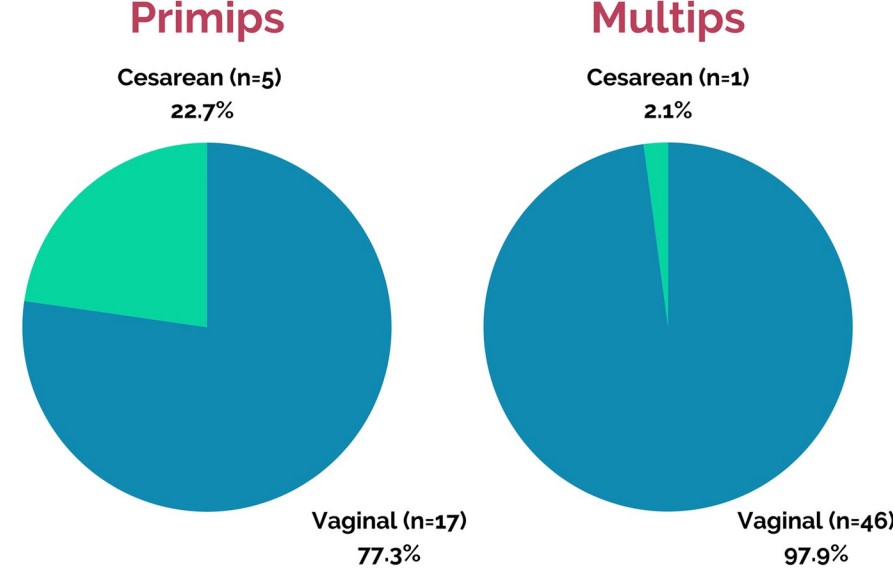

**Fig 2. Vaginal twin birth rates by parity.**

**Table 3. Type of presentations and vaginal birth rates.**

| Presentation | % | N | Vaginal birth rate |
|---|---|---|---|
| *Total* | *100%* | *69* | *91.3%* |
| Vertex/vertex | 50.7% | 35 | 97.1% (34/35) |
| Vertex/breech | 24.6% | 17 | 94.1% (16/17) |
| Vertex/transverse* | 4.3% | 3 | 66.7% (2/3) |
| Breech/vertex | 14.5% | 10 | 80% (8/10) |
| Breech/breech | 5.8% | 4 | 75% (3/4) |

* One converted spontaneously to vertex/vertex after baby A, another had an IPV (internal podalic version) and breech extraction due to separation of placenta A and heavy maternal bleeding. The third transported in labor and had a cesarean section.

incomplete, and 1 was footling. Of the 11 breech-first twins that birthed vaginally, 2 (18%) required more extensive maneuvers, such as freeing entangled heads and an extraction for prolapsed cord. Of the 19 breech second twins that birthed vaginally (not including the transverse lie that ended with IPV and extraction), 9 required more extensive maneuvers (fundal pressure, Pinard maneuver, and/or extraction) (47.4%).

For a description of the 8 in-labor transports, please see S3 File. We had one VBAC mother who began having suprapubic pain of a concerning nature at 10cm. She was transported via ambulance to a hospital, and no uterine scar dehiscence was found at repeat cesarean. She recovered without further sequelae.

## Induction vs. spontaneous labor

Our model of care supports waiting for natural labor to occur, given the benefits of spontaneous labor and the risks of induction. We did not routinely induce twins, which differs from

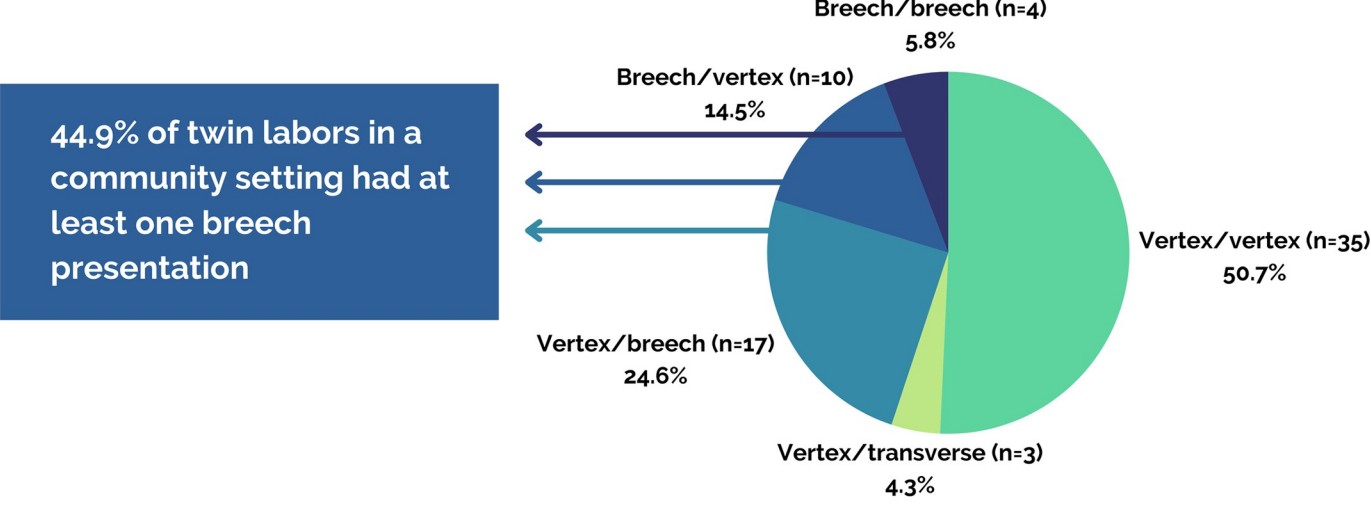

**Fig 3. Fetal presentations in twin labors.**

ACOG guidelines [28]; instead, we provided individualized care, weighing the risks and benefits that each mother thought suited her. Beginning at 39–40 weeks, SJF offered antepartum surveillance consisting of twice-weekly non-stress tests and biophysical profiles, which clients were free to accept or decline. Medical indications for induction resulted in an antepartum transfer of care (see Table 1). On a few occasions nearing 42 weeks, castor oil was used upon maternal request. Mean gestational age for home twin labor was 39.0 weeks (range 35–42 weeks).

### Twin-to-twin birth interval and estimated blood loss (EBL)

Overall, the twin-to-twin interval averaged 35.1 minutes. Early in the series, the birth team practiced less active intervention and waited for the spontaneous return of labor. When a trend appeared showing that longer inter-twin interval was related to increased blood loss, we began recommending AROM (artificial rupture of membranes) after 30–45 minutes if labor had not returned.

We have EBL data on 51/61 successful community births. Overall EBL average was 760.5 ml (range 150–1900). There were 16 twin mothers who had a EBL of ≥1000 ml (30.8%); of those 16, half had an EBL ≥1500 ml (15.3%). None of these women required transport or received a blood transfusion. Of the 16 mothers with an EBL of ≥1000 ml, we found the twin-to-twin interval to average 62 minutes (range 8–259) whereas in births with EBL <1000 ml, the interval averaged 24.7 minutes (range 2–135). Comparing mothers with an EBL above or below 1500 ml, the average twin-to-twin interval was 24 minutes for EBL <1500 ml (n = 43) and 85 minutes for EBL ≥1500 ml (n = 8). (See Fig 4).

During pregnancy, SJF focused on good nutrition and supplements to achieve maximal hemoglobin levels. Most women with excess blood loss received bimanual uterine massage, herbs (witch hazel tincture, shepherd's purse, and yunnan baiyao [a Chinese herb], all anti-hemorrhagic herbs used by the midwives that SFJ collaborates with), and homeopathy. In a few cases with a history of excessive blood loss, active management with IM Pitocin was given shortly after the birth of twin B. When clinically indicated, excessive bleeding was treated with IM Pitocin, IV fluids, IV Pitocin, rectal misoprostol and occasional IV tranexamic acid

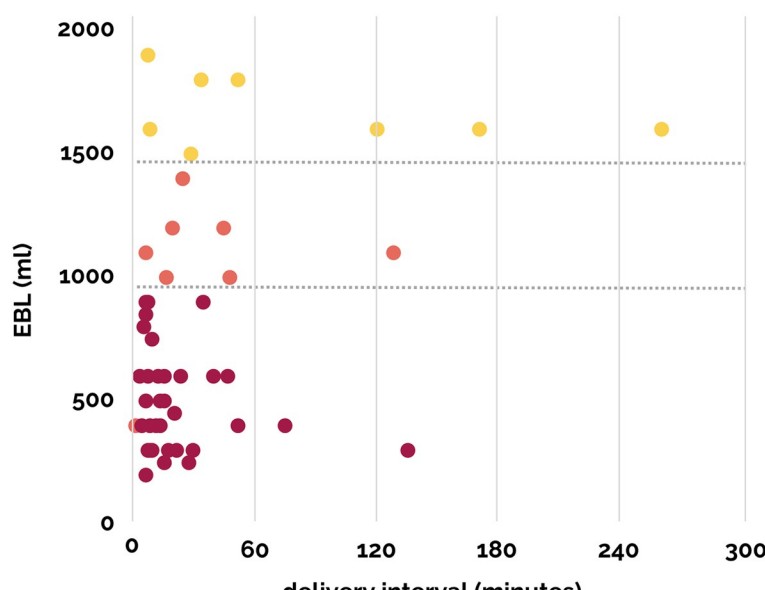

**Fig 4. Birth Interval and EBL.**

infusion. Early latching was encouraged to enhance uterine contractions. We do not have data on the length of the third stage of labor, but our practice was to wait for spontaneous separation and expulsion of the placenta unless intervention was clinically indicated. We did not have a structured time limit for the birth of the placenta.

## Perineal integrity

The majority of twin mothers birthing in a community setting had no perineal lacerations, with only one 3rd-degree laceration (repaired at home with lidocaine local anesthesia). As expected, perineal lacerations are more heavily skewed towards primiparous mothers. There was also one 3rd-degree laceration among a primip who transported in labor and had a spontaneous vaginal birth for baby A and vacuum delivery for baby B in a hospital. We do not have perineal outcomes for the other in-labor transport that ended in a vaginal birth. No episiotomies were performed in the community setting over a total of 122 vaginal births. See Figs 5 & 6 for perineal outcomes in completed twin community births.

The only maternal postpartum transport was for an eclamptic seizure which occurred about one hour postpartum. She recovered without further sequelae (S4 File).

## Neonatal outcomes

For deliveries that occurred in the community setting, the average birth weight of twin A was 2836 g (n = 61, range 1814–4337). The average birth weight of twin B was 2837 g (n = 61, range 1644–4224). Discordance between twins averaged 11.8% (n = 95, range 0–46%). Apgar scores at 1 minute averaged 8.0 for twin A (range 3–10) and 7.0 for twin B (range 3–9); this difference was statistically significant (p = 0.0005, unpaired t-test). At 5 minutes there was no significant difference in Apgars, with twin A averaging 9.0 (range 7–10) and twin B at 8.8 (range 6–10) (p = 0.13, unpaired t-test). There were two low 5-minute Apgars scores of 6, both among twin B. One had Goldenhar's syndrome. The other was after fetal bradycardia, leading to an IPV & breech extraction. Cord gasses were not collected in this setting.

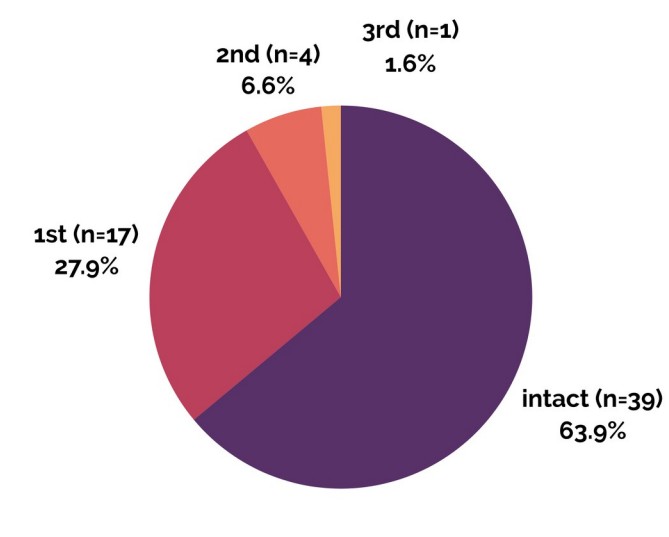

**Of the twin births completed at home, over 63% of women had intact perineums**

**Over 90% had little or no perineal trauma**

**Only one 3rd degree tear (1.6%)**

**No episiotomies**

**Fig 5. Perineal outcomes of vaginal twin births.**

We found no significant differences in 1 or 5 minute Apgar scores for twin B in relation to the birth interval, whether above or below 10 minutes, 30 minutes, or 60 minutes (unpaired t-test, p values were all > 0.05).

There was one non-emergent newborn transport of a twin A for persistent tachypnea and desaturation which resolved at the hospital within 48 hours and was attributed to be most likely transient tachypnea of the newborn (TTN, see S4 File). There was one case of short-term

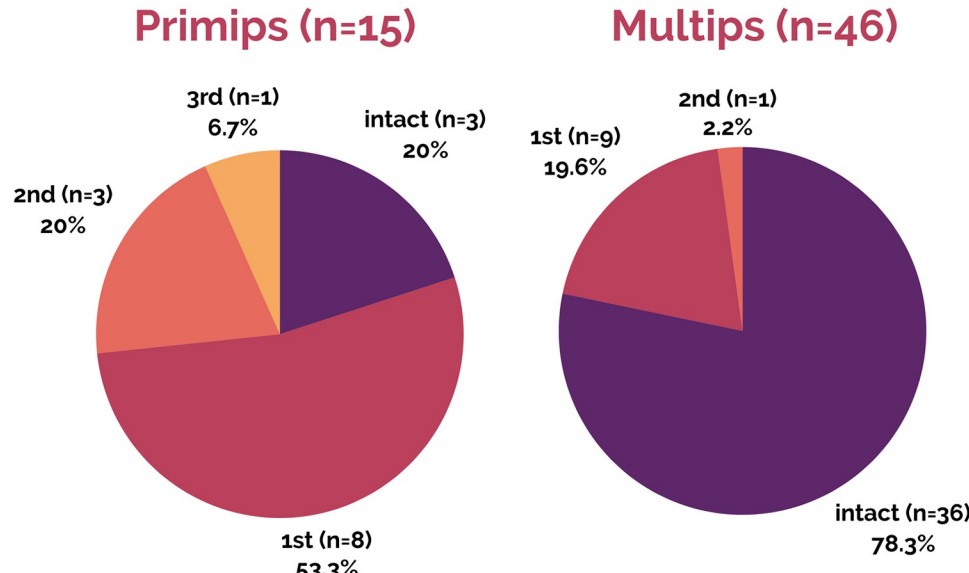

**Of the twin births completed at home, primip twin mothers had higher rates of perineal trauma compared to multips**

**Almost 3/4 of primips still had little or no perineal trauma**

**Fig 6. Perineal outcomes of vaginal twin births by parity.**

neonatal morbidity: a broken humerus on a second twin with a difficult breech extraction. The baby remained at home and healed without issue.

## IVF results

There were 15/100 women who had IVF. Four transferred care prior to labor. Three women with IVF had preterm labor or PPROM (21.4%) and one went to 41 weeks and chose a C-section. Of the 11 who began labor still under SJF's care, delivery by cesarean occurred in four (36.4%) with the remaining 7 having a vaginal birth, one after transport and 6 at home. Two women had intrauterine insemination (IUI) with both birthing vaginally at home.

## Discussion

In our study we found high rates of vaginal birth in the community setting of term twins, independent of chorionicity or fetal position. The overall vaginal birth rate was 91.3%. This was especially evident in multiparous women with a success rate of 97.9% for those still under SJF's care when labor began (46/47). Primiparas were less likely to birth vaginally but still had an overall rate of 77.3% (68.2% in the community setting and the remainder after hospital transport). Our vaginal birth rate was higher than the MANA Stats data by Bovbjerg at al., which had an 81.1% overall vaginal birth rate in a planned community setting [21]. The vaginal birth rate is markedly higher than large hospital-based studies such as the Twin Birth Trial [29], JUMODA [30], or the breech-first twin subset from JUMODA [31] (with the caveat that these are less direct comparisons than the MANA Stats data, see Fig 7 and Table 4).

Nine of the 61 births involved instrumental delivery, mostly for twin B. In a setting where instrumental delivery would not have been an option (for example, with midwives who cannot use forceps or vacuum), some of these births may have resolved without intervention. SJF is skilled and comfortable using instrumental delivery, and his threshold may be lower compared to other practitioners who cannot use these tools. Other may have likely transported earlier before the situation became emergent enough to perform an operative vaginal delivery. Some

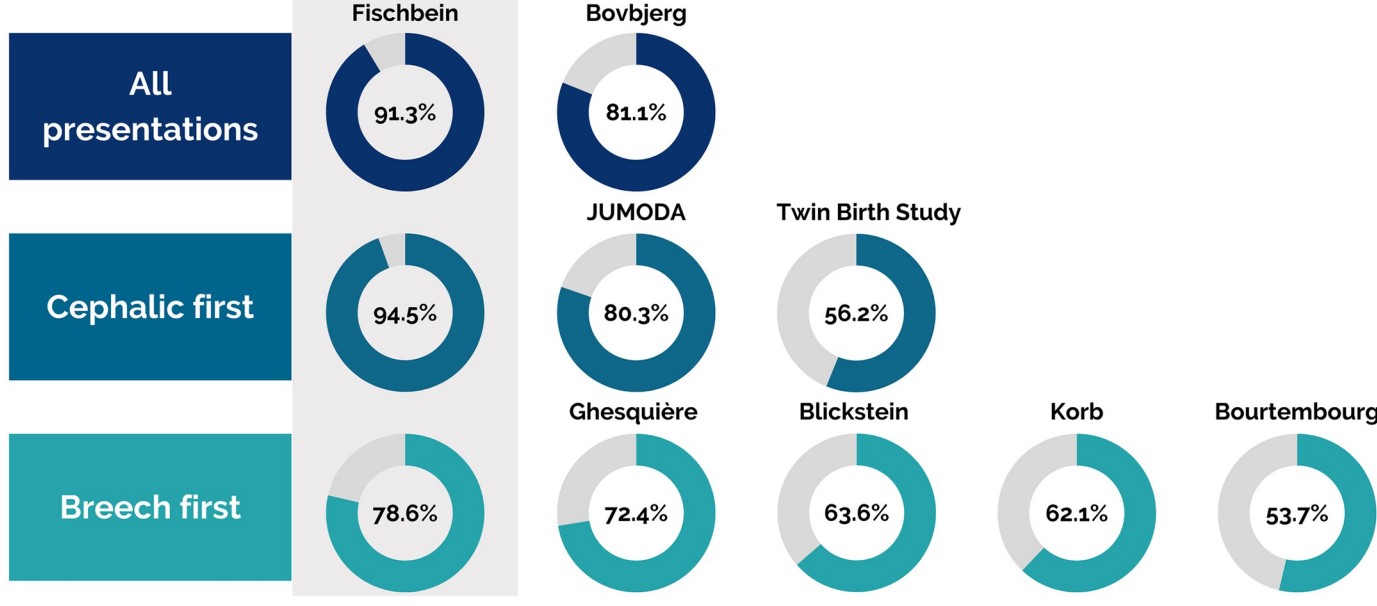

**Fig 7. Reported success rates of twin labors intending a vaginal birth.**

**Table 4. Comparison of vaginal twin birth rates.**

| Data set | N planned vaginal births | Presentation | Vaginal birth rate (n) | *primips (n)* | *multips (n)* |
|---|---|---|---|---|---|
| Fischbein | 69 | All | 92.3% (63) | *77.3% (16/22)* | *97.9% (46/47)* |
| Bovbjerg (MANA Stats 2.0 & 4.0) [21] | 138 | All | 81.1% (112) | *unspecified* | |
| **Cephalic-first twins** | | | | | |
| Fischbein subset | 55 | cephalic first | 94.5% (52) | *82.4% (14/17)* | *100% (38/38)* |
| Twin Birth Study (vaginal group) [29] | 1406 | cephalic first | 56.2% (790) | *unspecified: cohort was 61.5% multips* | |
| JUMODA planned vaginal group [30] | 5915 | cephalic first | 80.3% (3583) | *unspecified* | |
| **Breech-first twins** | | | | | |
| Fischbein subset | 14 | breech first | 78.6% (11) | *60% (3/5)* | *88.9% (8/9)* |
| Korb 2020 [32] | 298 | breech first | 62.1% (185) | | |
| Bourtembourg 2012 [34] | 54 | breech first | 53.7% | *32%* | *68.8%* |
| Blickstein 2000: at least 1500g [33] | 286 | breech first | 63.6% | *n = 53 Rate unspecified* | *n = 129 Rate unspecified* |
| Ghesquière 2022 [35] | 116 | Breech first | 72.4% (84) | *62.0% (31/50)* | *80.3% (53/66)* |

of these situations could also have been managed by IPV and breech extraction. SJF feels that vacuum delivery is a skill that midwives could easily learn, if allowed; however, governmental regulations and midwifery scope of practice laws generally do not support this skill training. Due to the availability of instrumental delivery in a home setting, the results of this study may not be directly applicable to the current state of midwifery practice guidelines.

During consultations, teaching, and/or clinical practice, the authors have noted that most women are told a cesarean is best when one or both of the twins are breech or when twin B is in an unstable lie. This has more to do with lack of expertise or willingness on the part of practitioners rather than evidence showing poorer outcomes with vaginal births [32–35]. For example, Korb et al. [32] analyzed outcomes of breech-first twins and commented: "despite the lack of solid evidence, planned vaginal delivery [for breech-first twins] has been progressively abandoned, resulting, as in the delivery of breech singletons, in a loss of expertise in the delivery of twin pregnancies with a first twin in breech presentation." Commenting on Korb's analysis, Aviram et al. [36] wrote in 2022: "a more recent publication based on the JUMODA dataset suggested that there is no difference in neonatal outcomes between planned CD and planned VD in twin pregnancies in which the first twin is in breech presentation. However, given the worldwide decrease in expertise in singleton breech deliveries, it is unlikely that this publication will change current practices."

We call this the "shrug" effect: an observation of a problem but no concerted efforts to fix it despite well-supported evidence. In essence, practitioners and hospital administrators shrug their shoulders and allow the overuse of cesareans to continue. We strongly encourage all birth attendants—whether obstetricians, family practice physicians, or midwives—to seek out breech skills training and mentorship. This may not always be included in residency programs and may need to be sought out independently via programs offered by Breech Without Borders, PROMPT, ALSO, BABE, Reteach Breech, and other organizations. Hospital programs have seen a rapid uptake of vaginal breech birth among women with breech presentations after staff have sought training and then opened a vaginal breech program. For example, the Southern Hospital of Denmark in Aabenraa brought physiological breech training to their entire team, promoted it as an option within the community, and saw planned vaginal breech births rise from around 50% in 2014 to over 80% by 2018 [41].

We urge practitioners to abandon the current obstetric practice of counseling women that breech-first twins are too dangerous to be delivered vaginally. The absence of data showing better outcomes with cesarean section for breech-first twins should be clearly communicated to both practitioners and parents. As Korb et al. note in 2020 [32]: "the absence of neonatal benefits associated with planned cesarean section in this study further emphasizes the reported increase in short-term and long-term maternal and infant risks associated with planned cesarean deliveries." [See, for example 42–48]. Twin pregnancies seem to have a different proportion of breech presentations compared to singleton breeches. While approximately two-thirds of term singleton breeches present frank breech [49], nonfrank breeches in our twin sample were much more common (67.6%) than frank breeches (32.4%).

Vaginal breech birth of twin A or B and extraction of twin B, when indicated, should be core obstetric skills taught to all obstetric and family practice residents as well as midwives. On average, more than half of twin pregnancies involve at least one breech presentation: 53% of twin pregnancies in a 2009 study [50]. Obstetricians currently in practice should seek out breech training to ensure they can safely support all twin births and uphold maternal autonomy [51]. Midwives have shown a much keener interest in retaining this skill as represented by their high rates of attendance in vaginal breech trainings conducted throughout the USA and Canada beginning in 2019 [52].

In our population, 31 (44.9%) of the women in labor had at least one twin in a breech position and 87.1% (27/31) of those had a successful community-based vaginal birth. There were 35 breech babies overall among the 31 mothers; 4 mothers had breech/breech twins. Of those 31 breech babies born with SJF, 35.5% (11/31) were assisted by breech extraction, 31% (10/31) were assisted with simple maneuvers often associated with non-upright positioning and 10 breech babies were born spontaneously (31%). One of the three transverse presentations also required internal version and breech extraction. This suggests the need for breech skills training for any practitioner attending vaginal twin births, due to the high percentage of breech presentations and to the occasional need for maneuvers, major or minor.

One set of breech/vertex twins had head entanglement, an extremely rare event estimated to occur in 1:1000 twin births overall [53]. In other studies, Korb [32] cites one case of interlocked twins out of 298 planned breech-first twins, while Blickstein [33] had no cases of entanglement among the 286 planned vaginal births of breech-first twins. The entanglement became evident after baby A was born to the chest with no further progress and unsuccessful attempts to maneuver the baby. Reaching inside past baby A's head and using his hand as a fulcrum, SJF was able to successfully unlock the two heads and deliver both babies promptly with good outcomes [54]. Knowing the mechanics of breech birth and having the skill and experience to understand what was happening, SJF was able to resolve a situation that is rarely encountered.

## Chorionicity

For Mono-Di twins, chorionicity should be determined as early as practical to outline a best plan of action and counseling due to the risk of developing TTTS & TAPS (twin anemia polycythemia sequence) [55]. Our Mono-Di twins began surveillance at 14–16 weeks with follow-up scans every 2 weeks until 28 weeks. SJF's experience is that TTTS becomes much less likely if there is no evidence by 28 weeks. However, TTTS can occur at any gestational age for Mono-Di twins [55], so ongoing monitoring for TTTS would be discussed with the parents. Four of our twin mothers developed TTTS over the period of our study. All four reached the level where fetal laser surgery was indicated between 20–25 weeks. All four had successful ablation of the communicating vessels and all eight babies survived. Three delivered by cesarean

from 29.5 to 33 weeks and one mother delivered vaginally (vertex/vertex) at the hospital at 33 5/7 weeks. With Di-Di twins we recommended growth scans every 4–6 weeks unless there were predisposing factors indicating need for more frequent surveillance. We encouraged all twin clients to have a 20-week structural survey ultrasound.

## To induce or not to induce?

The average gestational age at birth in our population of laboring women was 39 weeks (range 35–42 weeks). This differs significantly from how most hospitals manage twin pregnancies, with induction or C-section several weeks earlier than our population's average [27, 28]. The relative rate of stillbirth rises for twins beyond 35 weeks, hence the recommendation that all twins should be delivered by 37–38 weeks. (The relative rate of stillbirth rises for singletons beyond 35 weeks, as well). We raise this point to show that while the relative risk rises for both singletons and twins towards the end of pregnancy, that does not reflect the absolute risk, which is dependent on the denominator.

Our clients' preference for low intervention births was one of the main reasons they sought a community rather than a hospital birth. In our population we did not recommend induction simply based on gestational age. We believe that counseling based on relative risk alone is not helpful and even sometimes deceptive and that absolute risk should be discussed. This sets the risks in context of how common or rare they are. We also trust the evidence and predictive value of fetal surveillance [56] and generally follow ACOG guidelines when recommending its initiation after thorough informed consent counseling. During the informed consent process, we rely on the information from Page et al. [57], which shows the risk of PND/NND for twins at each gestational age.

In twins, excluding those complicated by gestational diabetes, hypertension, and IUGR, there is almost a 7-fold increase in the risk of IUFD between 37 and 40 weeks (relative risk). On the other hand, there is a 99.93% chance of not having an IUFD at 37 weeks and a 99.53% chance of not having an IUFD at 40 weeks (absolute risk). When counseling women, practitioners should explain the risks in an unbiased way and state absolute, not relative risk. The American Medical Association (AMA) code of ethics states, "Rational, informed patients should not be expected to act uniformly, even under similar circumstances, in agreeing to or refusing treatment" [58]. It is a basic tenet of medical ethics that when people are given the same information, they will not all reach the same conclusions [59]. It is not surprising then to understand why many in our population were willing to wait for spontaneous labor despite the small increase in the absolute risk of stillbirth. For our population, the risks of induction or cesarean section outweighed the risks of awaiting labor. We did offer (but not require) fetal biophysical profile testing beginning in the 38th week to reassure the family, the team and the potential hospital transport practitioner.

## Care during labor

Care during twin labors in a community setting differs significantly from how most twins are managed in a hospital setting [28, 60]. The care practices that our population finds objectionable include moving the mother to an operating suite for a vaginal birth, having a room full of staff, most of whom are strangers to the woman; lithotomy positioning; coached pushing; epidurals (often mandatory); immediate cord clamping; separation of mother and newborn; and AROM and immediate delivery of twin B within just a few minutes of twin A. In contrast, we support the mother in a safe, private environment where she is surrounded by people she knows. She is free to move at will, to eat and drink, to birth in whichever positions she finds most comfortable, most often to await the birth of baby B, and to have uninterrupted skin-to-

skin contact with her baby after the birth. These are all normal practices of the midwifery model of care and promote undisturbed births [37–40].

**Birth interval.**   We believe that a short, actively-managed birth interval is a routine born of impatience and convenience. For example, the average birth interval in the JUMODA study was 5 minutes [30] We acknowledge that the literature shows some associations between birth interval and lower cord gasses in twin B [61, 62]. However, we also see the benefits to letting the labor unfold without intervening for a specific indication. Hastening the birth of twin B should be only done in the rare indications where immediate interventions are medically indicated (such as fetal bradycardia, excessive bleeding, or cord prolapse). Women should be free to move about and push and deliver in whatever position feels best to them.

**Optimal cord clamping.**   Babies should have immediate skin to skin and optimal cord clamping (defined as waiting until the cord is limp, white, and flat before clamping & cutting the cord). "Immediate, continuous, uninterrupted" skin-to-skin contact is recommended by both WHO and Unicef for all babies >1000g and after all modes of birth because it improves both short- and long-term outcomes for both mother and baby [63, 64]. The Cochrane review, which included 38 relevant studies, agrees that "the evidence from this updated review supports using immediate or early SSC (skin-to-skin contact) to promote breastfeeding" [65]. In the immediate period after birth, immediate cord clamping (ICC) reduces blood volume by 25–30% in term infants and 50% in preterm infants [66, 67]; leads to a higher need for resuscitation [68–71]; and higher rates of hypoxic brain injury [72]. A randomized controlled trial of depressed infants born vaginally (non-breathing and in need of resuscitation) found that resuscitation with the cord intact, versus with ICC, led to improved Sp02 (pulse oximetry) levels and improved Apgar scores with no negative consequences [73]. Immediate cord clamping also leads to long-term problems such as iron deficiency anemia [74–78]; lower hemoglobin and hematocrit levels [74, 79], lower ferritin levels for six months [74], loss of stem cells which may lead to long-term health problems [80], and reduced social and fine motor skills at age 4 [81]. For more information on optimal cord clamping, we recommend Wait For White (waitforwhite.com).

In Mono-Di twins with no evidence of TTTS, we saw no reason to alter our practice of optimal cord clamping. Baby A can be put to breast to help stimulate contractions for Baby B. We recommend respectful, silent waiting for at least 30 minutes, intermittently monitoring Baby B's heartbeat, to let the mother and her uterus do what nature intends before suggesting an intervention. Like so much in obstetrics, these routine habits are done without concern for downstream consequences and the desires of the mother.

**Third stage management.**   In women with no risk factors for postpartum hemorrhage, we discussed but did not demand active third stage management (immediate injection of synthetic oxytocin, early cord clamping, and controlled cord traction). They were given informed consent about the increased risk of blood loss after a twin delivery and their decision to accept or decline routine postpartum Pitocin (synthetic oxytocin or Syntocinon) injection was respected. While some women were accepting of immediate IM Pitocin after delivery of twin B, many of our mothers preferred to avoid pharmaceuticals, which is common in the community birthing population.

EBL levels in our population are higher than in reported hospital studies, but with lower rates of blood transfusion. For example, in the Twin Birth Study [29], 2.3% of the planned vaginal births had EBL ≥1500 ml and 5.4% received blood transfusions, while our cohort had a rate of EBL ≥1500 ml of 15.3% but with no blood transfusions needed. Some of this may be complicated by the accuracy of estimating blood loss, and some may in part be our population's preference for physiological third stage and for longer average delivery interval times (the JUMODA study interval averaged 5 minutes in the planned vaginal birth group [30],

while our population averaged 35.1 minutes). Our series of births had no transfusions because the midwifery model of care allowed SJF to keep the mothers stable, even after blood loss, by one-on-one care, vigilant monitoring, and other treatments such as IV fluids. None of these women were unstable enough to warrant transfer for transfusion. The entire birth team stayed on average 4–6 hours postpartum and often families hired an allied birthworker (doula, lactation consultant, etc.) to stay in the home for continued support.

## Limitations

While our dataset has detailed information on the pregnancies and births, it is too small to accurately calculate the incidence of rare events, such as severe neonatal morbidity or mortality. We are aware of home birth practitioners in the USA who have attended over 500 sets of twins, but their data has never been analyzed. For example, Cynthia Caillagh, a Certified Professional Midwife in rural Wisconsin, attended 561 vaginal breech births and 527 sets of twins in a community setting over her 50-year career as a midwife. In the past decade, she had a >97% vaginal birth rate for her breech and twin births (personal correspondence with RF). She was compiling her breech and twin data for RF to publish when she died of cancer. We also call for similar studies to be done around the world to understand twin birth outside of an American context. We strongly encourage all practitioners, whether community- or hospital-based, to collect and publish their data on planned vaginal births of twins.

## Conclusions

Properly selected, motivated twin mothers can successfully birth in a home setting, if they have a supportive environment, a skilled practitioner, and a competent team of birth workers. Properly selected term twin vaginal birth, regardless of which twin is breech, is a reasonable option with excellent success rates in skilled hands and should be offered to all women with twin pregnancies. Residency training programs should reinstitute the teaching of these skills to ensure that women desiring hospital births are not pushed outside the system. Professional medical organizations and state legislatures should recognize the value and dedication of midwifery care in the community and stop limiting choices, under the guise of safety, that rightfully belong to the pregnant woman.

The midwifery approach to term twin pregnancy is significantly different from the obstetric medical model, and women seek it out despite obstacles of cost, relocation, and social disapproval. Most twin mothers entered into our practice early in gestation, but some transferred or relocated later in the pregnancy, usually due to lack of birth options in their community. The midwifery approach supports physiology and strongly supports vaginal birth in the context of the woman's individual values and preferences. A medicalized approach to multiples sees twin birth as hazardous and relies heavily on algorithmic managing of the prenatal period, following a high-risk protocol of consults and ultrasound, and then managing the birth process via cesarean section or induction of labor. In SJF's experience attending births for over four decades in both hospital and community settings, this approach causes anxiety and fear in the women and their families, with the anxiety often originating from the practitioner. In contrast, when a woman is confident and feels supported, her prenatal experience and labor are more enjoyable. While alertness and vigilance are appropriate, a fear-based defensive approach to twins is unwarranted and detrimental.

As opposed to the pre-ultrasound era, when half of all twins were discovered in labor, most women discover a twin pregnancy early on. We know that around half of all twins will have at least one baby in the breech position [50]. Many obstetricians have no training or experience in vaginal breech birth; breech extraction may be the only skill they learn and the only tool

they have, limited to twin B. Without vaginal breech skills, practitioners cannot be considered experts in twin birth because they lack the skills for an event (breech presentation) that occurs almost half of the time.

We argue that obstetricians should not be taking care of twins if they cannot or will not support vaginal breech births, since roughly half of all twins involve one or more breech presentations. However, they continue to do so and rarely refer their twin clients to someone skilled and confident in twin vaginal delivery. But this is exactly what must be done if we are to honor our code of ethics as maternity practitioners.

We also propose some common sense suggestions for less experienced practitioners attending twin births:

- Seek out breech training and practice the associated maneuvers monthly

- Adhere to reasonable selection criteria, honoring one's personal comfort levels

- Gestational age of 35 weeks or greater (in a community setting)

- Multiparous mothers

- Diamniotic/dichorionic twins and relative concordance (due to a need for more frequent surveillance and consultation with discordant growth and Mono-Di twins)

As skill and comfort levels increase, practitioners can offer more expansive options such as primiparas and uncomplicated Mono-Di twins.

Instead of creating an ever-expanding list of contraindications to home and birth center births—including twins—we propose that we *contraindicate restrictions on maternal choice of practitioner or birth setting* and *contraindicate restrictions that limit providers in both community and hospital settings from supporting vaginal twin or breech births*. Contraindicating certain choices is paternalistic, assuming the doctor, obstetric society, or regulatory committee knows better than the woman herself. It also ignores individual values, beliefs, and preferences, seeking to impose a one-size-fits-all maternity care experience that may benefit some while harming others. The goal should be to create a nurturing environment, where each woman is treated with dignity, compassion, and respect by a skilled twin practitioner and where her choices are not just tolerated but enthusiastically supported.

## Supporting information

**S1 Data. Spreadsheet of deidentified twin home birth data.**
(XLSX)

**S1 File. Video of a twin community birth with SJF.** This file includes a video link to a twin birth attended by SJF.
(DOCX)

**S2 File. Details on VBAC twin labors.** This file includes details on the six VBAC twin labors.
(DOCX)

**S3 File. History of 8 twin labor transports.** This file describes the circumstances surrounding the eight in-labor transports, of which one was considered emergent.
(DOCX)

**S4 File. Details on neonatal & maternal transports.** This file describes the two postpartum transports: one neonatal transport for cyanosis and persistent tachypnea and one maternal

transport for seizures.
(DOCX)

## Acknowledgments

Our paper would not have been possible without the dedication and collaboration of the Southern California community based midwives, doulas, students and other birth workers who supported the author and the families throughout the pregnancy, labor and postpartum. It is with great appreciation and gratitude we acknowledge their contribution. We also hold deep respect for the women and families who chose a different path and placed their trust in SJF's team.

## Author Contributions

**Conceptualization:** Stuart J. Fischbein.

**Data curation:** Stuart J. Fischbein.

**Investigation:** Stuart J. Fischbein.

**Methodology:** Stuart J. Fischbein.

**Validation:** Rixa Freeze.

**Visualization:** Rixa Freeze.

**Writing – original draft:** Stuart J. Fischbein, Rixa Freeze.

**Writing – review & editing:** Stuart J. Fischbein, Rixa Freeze.

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
