## [Decision Letter · Decision Letter 0]

5 Aug 2024

PONE-D-24-05781Twin home birth: Outcomes of 100 sets of twins in the care of a single practitionerPLOS ONE

Dear Dr. Freeze,

Thank you for submitting your manuscript to PLOS ONE. After careful consideration, we feel that it has merit but does not fully meet PLOS ONE’s publication criteria as it currently stands. Therefore, we invite you to submit a revised version of the manuscript that addresses the points raised during the review process.

We look forward to receiving your revised manuscript.

Kind regards,

Tamara Sljivancanin Jakovljevic

Academic Editor

PLOS ONE

Reviewers' comments:

Reviewer's Responses to Questions

**Comments to the Author**

1. Is the manuscript technically sound, and do the data support the conclusions?

Reviewer #1: Partly

Reviewer #2: Yes

2. Has the statistical analysis been performed appropriately and rigorously? 

Reviewer #1: Yes

Reviewer #2: No

3. Have the authors made all data underlying the findings in their manuscript fully available?

Reviewer #1: No

Reviewer #2: Yes

4. Is the manuscript presented in an intelligible fashion and written in standard English?

Reviewer #1: Yes

Reviewer #2: Yes

5. Review Comments to the Author

Reviewer #1: Dear authors,

I have reviewed your manuscript reporting outcome of 100 twin homebirths. You have made a great effort presenting a significant number of twin home deliveries during 12 -year period. Your described management of home twin births in detail and presented results in seven figures and four tables with four supplement supporting information that included one video, two word documents and an excel table.

However, as you mentioned in your work, this is rather controversial topic, as it opposes standard protocols, especially ACOG. Therefore, many questions arise concerning mother and children safety during home birth, especially in women with previous Cesarean section (CS) and first baby breach. You must agree that those situations present risk for both mother and the baby even in hospital settings and home birth in those cases may be considered hazardous. I presume that you have informed your patients of all possible risks they are exposing and that you have an excellent insurance in the cases of potential adverse event and low suit.

You delivered both dichorionic and monochorionic-diamniotic twins. There is an ongoing debate about timing of delivery in monochorionic-diamniotic twins as potential stillbirth may occur in those cases. Monitoring those pregnancies demands a perinatology specialist and we may say that monochorionic-diamniotic pregnancy was uneventful only after delivery. How did you monitor monochorionic-diamniotic twins concerning TAPS (Twin Anemia Polycythemia Sequence) and did any of monochorionic-diamniotic twins have TAPS diagnosed during pregnancy or after delivery? How did you monitor those pregnancies after 38 weeks, as 37/38 weeks is the choice for delivery by ACOG protocols?

VBAC is an excellent choice for all uncomplicated pregnancies and deliveries, and should be conducted under intensive monitoring in hospital setting. Concerning patients with previous CS, what was their parity, i.e. did they have previous vaginal delivery as well, and was vaginal delivery prior or after CS? What were indications for previous CS?

All the patients were divided to primiparas and multiparas. It is questionable whether women with previous SC can be considered in the same way as women having their first delivery. I suggest you to make three groups, extracting women with VBAC in the separate group.

You mentioned that pregnancies were uncomplicated with eutrophic growth of the twins, but neonatal birth weight was from 1814g for the twin A and 1644g for the twin B, which is a low birth weight for term pregnancy even for twins.

As for the Apgar score, the lowest 1-minute Apgar was 3 for both twins, and 5-minute 7 for twin A, and 6 for twin B. How many neonates had low Apgar score, what was the reason and what resuscitation measures you performed in the home setting?

It is obvious that your patients opted not to have epidural anesthesia, but how did you managed the cases of retained placenta or excessive hemorrhage and the need of manual revision of uterine cavity? Did you perform it without anesthesia?

How did you monitor uterine integrity after VBAC?

What is average time needed for hospital transfer? Was there any problems concerning it?

Have you performed GBS screening?

In introduction, line 87, and in discussion you pointed that most transfers to the hospital ended in CS. In my opinion there is no need for to emphasize it in an almost negative way, since transfer had been made in risky situations in the cases when vaginal delivery was questionable.

There are many questions concerning home birth of a risk pregnancy and this topic is rather controversial. I encourage authors to answer the raised questions.

Reviewer #2: I would like to thank the authors for addressing this topic in obstetrics on a community-based level which is rarly done before. The results of the study will give a guide for management of such cases.

6. PLOS authors have the option to publish the peer review history of their article (what does this mean?). If published, this will include your full peer review and any attached files.

Reviewer #1: No

Reviewer #2: **Yes: **Mohsen M A Abdelhafez

---

## [Author Response · Author response to Decision Letter 0]

6 Sep 2024

Responses to Reviewer #1: 

"However, as you mentioned in your work, this is rather controversial topic, as it opposes standard protocols, especially ACOG. Therefore, many questions arise concerning mother and children safety during home birth, especially in women with previous Cesarean section (CS) and first baby breach. You must agree that those situations present risk for both mother and the baby even in hospital settings and home birth in those cases may be considered hazardous. I presume that you have informed your patients of all possible risks they are exposing and that you have an excellent insurance in the cases of potential adverse event and low suit."

Authors’ response: Our paper demonstrates that VBAC with twins and vaginal birth of breech-first twins is a reasonable option, both in hospital settings (as per our reference list mentioned in the paper) as well as in a home setting. SJF’s patient counseling is extensive, as his model of care allows him the time to review all risks, benefits, and alternatives to all possible options. His clients are well-informed and choose to reject the obstetrical model and ACOG guidelines because those do not align with their values. The discussion of liability insurance is irrelevant to this presentation of clinical data outcomes and shows that this reviewer is not familiar with common home birth practices; many home birth attendants in the US forego liability insurance. (This comment above does not seem to ask for revision in the paper itself, so we have addressed the comments in this author response.)

"You delivered both dichorionic and monochorionic-diamniotic twins. There is an ongoing debate about timing of delivery in monochorionic-diamniotic twins as potential stillbirth may occur in those cases. Monitoring those pregnancies demands a perinatology specialist and we may say that monochorionic-diamniotic pregnancy was uneventful only after delivery. How did you monitor monochorionic-diamniotic twins concerning TAPS (Twin Anemia Polycythemia Sequence) and did any of monochorionic-diamniotic twins have TAPS diagnosed during pregnancy or after delivery? How did you monitor those pregnancies after 38 weeks, as 37/38 weeks is the choice for delivery by ACOG protocols?"

Authors’ response: Our paper is part of this ongoing debate, showing that routine induction at a set time point may not be necessary, especially for those who are motivated to avoid induction (as is the case with our population). We disagree that mono-di pregnancies “demand” a perinatology specialist. SJF consulted with a perinatologist when appropriate but not in every case of mono-di twins. In lines 144- 152, we explain the monitoring protocols for mono-di pregnancies. SJF did his own ultrasound scans, allowing him to screen and monitor his twin cases directly for possible anomalies and complications. None of the mono-di twins developed TAPS or signs associated with TAPS. SJF did not screen specifically for TAPS, as TAPS alone would not have been an indication to change his management. As mentioned in the paper, SJF offered twice-weekly fetal surveillance after 38 weeks, which clients were free to accept or decline. Again, the model of care in a home birth setting allows for individualized care and patient autonomy; most patient in his model are looking for something outside of ACOG protocols. Please refer to lines 282-287 and 451-482 for our discussion of these issues. 

"VBAC is an excellent choice for all uncomplicated pregnancies and deliveries, and should be conducted under intensive monitoring in hospital setting. Concerning patients with previous CS, what was their parity, i.e. did they have previous vaginal delivery as well, and was vaginal delivery prior or after CS? What were indications for previous CS?"

Authors’ response: The statement that VBAC “should be conducted under intensive monitoring in hospital setting” is the reviewer’s belief and an ACOG recommendation, but again, we are examining a different style of care and a different approach to twins. Both SJF and the clients he serves desire an alternative approach to ACOG management; the paper demonstrates excellent outcomes without having to adhere to every ACOG guideline for twins. SJF did not take into account indications for previous C-sections as a limiting factor for who would plan a VBAC with twins, because there is no accurate way to predict the success of a VBAC based on indications for the previous surgery. Thus we have not included indications for prior C-sections in our data, as it was not clinically relevant to SJF’s care. In the Excel spreadsheet provided with the paper, you can see how many VBACs entering labor still under SJF’s care had prior vaginal births and (1) and how many were functional primips (no previous vaginal births, n=5). We have also added an additional supplemental file (S2) with more information about the VBAC twin births.

"All the patients were divided to primiparas and multiparas. It is questionable whether women with previous SC can be considered in the same way as women having their first delivery. I suggest you to make three groups, extracting women with VBAC in the separate group."

Authors’ response: To remain consistent with our prior publication of SJF’s breech & cephalic singleton data (Fischbein & Freeze 2018, cited in the paper), we would prefer to keep the two original groups. However, to address this reviewer’s concerns, we have included an additional addendum with more information about each VBAC (which is also available in the data spreadsheet attached with the paper). We feel that it is most appropriate to include functional primips (i.e., VBACs with no previous vaginal births) in the group with primips, while keeping VBACS with previous vaginal births in the multip group, as those two groups most closely align in terms of their risk factors and labor patterns. 

"You mentioned that pregnancies were uncomplicated with eutrophic growth of the twins, but neonatal birth weight was from 1814g for the twin A and 1644g for the twin B, which is a low birth weight for term pregnancy even for twins."

Authors’ response: While small, these twins were both growing well on their respective growth curves. As mentioned in the paper, SJF’s selection criteria included having both twins growing consistently on their own ultrasound growth curves, which they were even for these two smaller babies. 

"As for the Apgar score, the lowest 1-minute Apgar was 3 for both twins, and 5-minute 7 for twin A, and 6 for twin B. How many neonates had low Apgar score, what was the reason and what resuscitation measures you performed in the home setting?"

Authors’ response: 1-minute Apgar scores are generally not considered clinically relevant; most papers only publish 5-minute scores. We chose to mention both, but to respond to this reviewer’s question, there were two 5-minute Apgars of 6 for twin B and none for twin A. One twin B with a low Apgar score had Goldenhar’s syndrome. The other low Apgar for a Twin B was after fetal bradycardia, leading to an IPV & breech extraction. SJF’s team followed NRP resuscitation guidelines, leaving the cord intact during resuscitation. We added a description of the equipment carried as well as these details about the two cases of low Apgar scores.

"It is obvious that your patients opted not to have epidural anesthesia, but how did you managed the cases of retained placenta or excessive hemorrhage and the need of manual revision of uterine cavity? Did you perform it without anesthesia?"

Authors’ response: Yes, all of these situations were managed at home without anesthesia. 

"How did you monitor uterine integrity after VBAC?"

Authors’ response: If the mother was stable, SJF managed VBACs like any other postpartum patient. No additional uterine monitoring or exam was considered necessary. 

"What is average time needed for hospital transfer? Was there any problems concerning it?"

Authors’ response: We had no problems arise due to time of hospital transport among the two cases that transported via ambulance (maternal seizure postpartum and a laboring VBAC mother at 10 cm). The other case that transported (neonatal transport for persistent low 02 sats) was not urgent and transported via private vehicle. Los Angeles is a well-served urban area with ample access to EMS services and hospitals. On average, SJF’s clients lived under 30 minutes away from the nearest hospital and often much closer. 

"Have you performed GBS screening?"

Authors’ response: GBS screening was offered but not required, as with all potential screenings and interventions in SJF’s model of care. 

"In introduction, line 87, and in discussion you pointed that most transfers to the hospital ended in CS. In my opinion there is no need for to emphasize it in an almost negative way, since transfer had been made in risky situations in the cases when vaginal delivery was questionable."

Authors’ response: Since the reviewer was clear that this was a matter of personal opinion (“in my opinion”) and not an issue with the presentation of the clinical data, we are choosing to leave these parts as-is. We do not feel that including C-section rates after transport is biased or negative; rather, we are simply stating the outcomes of transports. We disagree that this data is presented in an “almost negative” way; it is simply presented as a matter of fact and percentage. Given that 2 of the 8 transports ended vaginally, and that more likely could have ended vaginally had vaginal birth been an option at the receiving hospital, whether the transports ended vaginally is relevant and important information. (In line 87, we were citing another article, not our own data.)

"There are many questions concerning home birth of a risk pregnancy and this topic is rather controversial. I encourage authors to answer the raised questions."

Responses to Reviewer #2:

"I would like to thank the authors for addressing this topic in obstetrics on a community-based level which is rarely done before. The results of the study will give a guide for management of such cases."

Authors’ response: Thank you

General comments from the authors:

Responding to one of the reviewer’s responses to the question: “Has the statistical analysis been performed appropriately and rigorously?”

• We added a note in the paper that the sample sizes were too small for most statistical analysis. 

Responding to one of the reviewer’s responses to the question: “Have the authors made all data underlying the findings in their manuscript fully available?”

• One of the reviewers responded “no.” We have made all data fully available via the included Excel spreadsheet. We cannot, of course, include the original medical charts with the patient’s name and identifying information, but we excerpted all relevant anonymized data onto the spreadsheet.

---

## [Decision Letter · Decision Letter 1]

24 Oct 2024

PONE-D-24-05781R1Twin home birth: Outcomes of 100 sets of twins in the care of a single practitionerPLOS ONE

Dear Dr. Freeze,

Thank you for submitting your manuscript to PLOS ONE. After careful consideration, we feel that it has merit but does not fully meet PLOS ONE’s publication criteria as it currently stands. Therefore, we invite you to submit a revised version of the manuscript that addresses the points raised during the review process.

We look forward to receiving your revised manuscript.

Kind regards,

Tamara Sljivancanin Jakovljevic

Academic Editor

PLOS ONE

Journal Requirements:

Reviewers' comments:

Reviewer's Responses to Questions

**Comments to the Author**

1. If the authors have adequately addressed your comments raised in a previous round of review and you feel that this manuscript is now acceptable for publication, you may indicate that here to bypass the “Comments to the Author” section, enter your conflict of interest statement in the “Confidential to Editor” section, and submit your "Accept" recommendation.

Reviewer #3: (No Response)

2. Is the manuscript technically sound, and do the data support the conclusions?

Reviewer #3: (No Response)

3. Has the statistical analysis been performed appropriately and rigorously? 

Reviewer #3: (No Response)

4. Have the authors made all data underlying the findings in their manuscript fully available?

Reviewer #3: (No Response)

5. Is the manuscript presented in an intelligible fashion and written in standard English?

Reviewer #3: Yes

6. Review Comments to the Author

Reviewer #3: ABSTRACT: The statement in lines 23-25 should be recasted and grammar corrected.

METHODS: How was the data recorded?

RESULTS: Under induction vs spontaneous labour, age 283-- how were the women whose pregnancies had passed their expected date of delivery manged?

Lines 301-302- for a statement of this nature, figures showing how significance was determined should be included.

DISCUSSION: In lines 316-317, was the 3rd degree laceration which was repaired at home repaired under anaesthesia in the house if so what type? In lies 368-369, the statement is not very clear, based on the results of their study which of the birth or indications of the instrumental vaginal deliveries are the authors referring to? Similarly, in lines 370-371, the authors should state for clarity which of the situations they would have managed otherwise and why they did not do so. The statement in lines 555-556 with quoted figures ideally should be referenced,

7. PLOS authors have the option to publish the peer review history of their article (what does this mean?). If published, this will include your full peer review and any attached files.

Reviewer #3: No

---

## [Author Response · Author response to Decision Letter 1]

30 Oct 2024

Reviewer #3: 

ABSTRACT: The statement in lines 23-25 should be recasted and grammar corrected.

• We corrected the grammar in these lines.

METHODS: How was the data recorded?

• We added further clarification on page 8, lines 217-218 to note that SJF excerpted the relevant information from the women’s medical records into an Excel spreadsheet. 

RESULTS: Under induction vs spontaneous labour, age 283-- how were the women whose pregnancies had passed their expected date of delivery manged?

• As noted in the paragraph under “induction vs. spontaneous labor”, SJF offered a different standard of care and did not mandate delivery by 37 or 38 weeks. Instead, after counseling and informed consent discussions, the normal practice was to await labor. We added another sentence in this section clarifying SJF’s normal practices for clients at or past 40 weeks. 

Lines 301-302- for a statement of this nature, figures showing how significance was determined should be included.

• The numbers were too small to calculate p-values; we have decided to omit this sentence to avoid confusion. 

DISCUSSION: In lines 316-317, was the 3rd degree laceration which was repaired at home repaired under anaesthesia in the house? If so what type?

• It was repaired with local anesthesia (lidocaine)

 In lines 368-369, the statement is not very clear, based on the results of their study which of the birth or indications of the instrumental vaginal deliveries are the authors referring to? Similarly, in lines 370-371, the authors should state for clarity which of the situations they would have managed otherwise and why they did not do so. 

• We added some clarification in the paper, noting that SJF’s use of instrumental delivery is related to his experience and skillset. Each case requires an individualized judgment call and is situational (mother’s exhaustion levels, anxiety in the room, etc.). It’s impossible to predict the outcomes of each specific birth with a different care provider who could not use instrumental delivery. We wish to stress that we cannot point to each individual birth and predict the specific outcomes had a midwife been present versus an obstetrician. However, SJF feels that, in general, a midwife may been able to resolve some situations in other ways and may have transported other situations earlier. 

The statement in lines 555-556 with quoted figures ideally should be referenced,

• If you are referring to Cynthia Caillagh’s data, it is currently unpublished and RF had personal correspondence with her prior to her death. There is not a formal reference available as her data is not yet published. We added a parenthetical citation to clarify this.

---

## [Editor Report · Decision Letter 2]

4 Nov 2024

Twin home birth: Outcomes of 100 sets of twins in the care of a single practitioner

PONE-D-24-05781R2

Dear Dr. Rixa Freeze,

We’re pleased to inform you that your manuscript has been judged scientifically suitable for publication and will be formally accepted for publication once it meets all outstanding technical requirements.

Kind regards,

Tamara Sljivancanin Jakovljevic

Academic Editor

PLOS ONE

---

## [Editor Report · Acceptance letter]

13 Nov 2024

PONE-D-24-05781R2 

PLOS ONE

Dear Dr. Freeze, 

I'm pleased to inform you that your manuscript has been deemed suitable for publication in PLOS ONE. Congratulations! Your manuscript is now being handed over to our production team.

Kind regards, 

on behalf of

Dr. Tamara Sljivancanin Jakovljevic 

Academic Editor

PLOS ONE